# Structural basis for recognition and remodeling of the TBP:DNA:NC2 complex by Mot1

Agata Butryn[1], Jan M Schuller[2], Gabriele Stoehr[1], Petra Runge-Wollmann[1], Friedrich Förster[2], David T Auble[3]*, Karl-Peter Hopfner[1,4]*

[1]Gene Center, Department of Biochemistry, Ludwig Maximilian University, Munich, Germany; [2]Department of Molecular Structural Biology, Max Planck Institute of Biochemistry, Martinsried, Germany; [3]Department of Biochemistry and Molecular Genetics, University of Virginia Health System, Charlottesville, United States; [4]Center for Integrated Protein Sciences, Ludwig Maximilian University, Munich, Germany

*For correspondence: auble@virginia.edu (DTA); hopfner@genzentrum.lmu.de (K-PH).

Competing interests: The authors declare that no competing interests exist.

**Abstract** Swi2/Snf2 ATPases remodel substrates such as nucleosomes and transcription complexes to control a wide range of DNA-associated processes, but detailed structural information on the ATP-dependent remodeling reactions is largely absent. The single subunit remodeler Mot1 (modifier of transcription 1) dissociates TATA box-binding protein (TBP):DNA complexes, offering a useful system to address the structural mechanisms of Swi2/Snf2 ATPases. Here, we report the crystal structure of the N-terminal domain of Mot1 in complex with TBP, DNA, and the transcription regulator negative cofactor 2 (NC2). Our data show that Mot1 reduces DNA:NC2 interactions and unbends DNA as compared to the TBP:DNA:NC2 state, suggesting that Mot1 primes TBP:NC2 displacement in an ATP-independent manner. Electron microscopy and cross-linking data suggest that the Swi2/Snf2 domain of Mot1 associates with the upstream DNA and the histone fold of NC2, thereby revealing parallels to some nucleosome remodelers. This study provides a structural framework for how a Swi2/Snf2 ATPase interacts with its substrate DNA:protein complex.

## Introduction

Swi2/Snf2 (switch/sucrose non-fermenting 2) ATPases form a large and diverse class of proteins and multiprotein assemblies within the SF2 superfamily of helicases/translocases (*Flaus et al., 2006*). Swi2/Snf2 enzymes are best known as the principal catalytic subunits of large chromatin remodeling complexes that regulate the spatial arrangement and histone composition of nucleosomes, e.g. SWI/SNF, ISWI, CHD, and INO80 remodelers (*Gangaraju and Bartholomew, 2007*; *Clapier and Cairns, 2009*). Current models propose that Swi2/Snf2 ATPases track the minor groove of DNA, thereby generating tension that induces transient or permanent alterations in molecular assemblies (*Saha et al., 2002*; *Whitehouse et al., 2003*; *Zofall et al., 2006*). Although high-resolution structures of some Swi2/Snf2 domains and their substrate-interacting domains associated with DNA and protein have been previously described, detailed structural information about these ATPases bound to substrate proteins and DNA is largely absent (*Dürr et al., 2005*; *Thomä et al., 2005*; *Shaw et al., 2008*; *Hauk et al., 2010*; *Sharma et al., 2011*; *Yamada et al., 2011*). The lack of structural insights is owed to the complex, often multisubunit architecture of Swi2/Snf2 enzyme-containing complexes and their inherent structural flexibilities.

Mot1 (modifier of transcription 1, also denoted BTAF1) is conserved from protozoa to humans and was the first Swi2/Snf2 member for which the biochemical activity has been demonstrated in vitro (*Auble and Hahn, 1993*). Mot1 dissociates TATA box-binding protein (TBP) from DNA in an ATP-dependent manner,

**eLife digest** An organism's DNA contains thousands of genes, not all of which are active at the same time. Cells use a number of methods to carefully control when particular genes are switched on or off. For example, proteins called transcription factors can activate a gene by binding to particular regions of DNA called promoters. One such transcription factor is called the TATA-binding protein (TBP for short). Mot1 is a remodeling enzyme that can form a "complex" with TBP by binding to it, and in doing so remove TBP from DNA. This silences the genes at those sites. The freed TBP can then bind to other promoters that lack Mot1 and activate the genes found there.

In 2011, researchers revealed the structure of the complex formed between TBP and Mot1 after TBP has been detached from DNA. However, the structure of the complex that forms while TBP is still bound to the DNA molecule was not known. Butryn et al. – including several of the researchers involved in the 2011 work – have now described the structure of this complex using X-ray crystallography and electron microscopy. Another protein called negative cofactor 2 is also part of the complex, and helps to stabilize it.

Butryn et al. found that Mot1 reduces the strength of the interactions between DNA and both TBP and negative cofactor 2. Binding to TBP and negative cofactor 2 causes the DNA molecule to bend; however, if Mot1 is also in the complex, the DNA becomes less bent. By making these changes, Mot1 is likely to prime TBP to detach from the DNA. Since the current structures do not yet reveal the atomic structure of Mot1's ATP dependent DNA motor domain, the next challenge is to visualize the entire complex at atomic resolution.

thereby directly regulating transcription initiation process and global redistribution of TBP in the cell (*Auble and Hahn, 1993*; *Dasgupta et al., 2002*; *Darst et al., 2003*; *Zentner and Henikoff, 2013*). In contrast to the majority of other Swi2/Snf2 enzymes, Mot1 does not require any associated subunit for its essential activity and serves as a useful model system for studying Swi2/Snf2 enzymes in vitro. The crystal structure of the N-terminal domain of *Encephalitozoon cuniculi* Mot1 (Mot1$^{NTD}$) in complex with TBP showed that Mot1 consists of 16 HEAT repeats (Huntingtin, elongation factor 3, protein phosphatase 2A, lipid kinase TOR) that are arranged in a horseshoe-like shape (*Wollmann et al., 2011*). Of note was that a long loop between HEAT repeats 2 and 3, denoted as the 'latch', can bind to TBP's concave site and block TBP–DNA association. Therefore, previous analyses have revealed not only how Mot1 binds TBP, but also that Mot1 functions as a TBP chaperone. Numerous biochemical and more recent structural studies of the Mot1:TBP complex predicted the approximate positioning of the ATPase domain (Mot1$^{CTD}$) with respect to the DNA upstream of the TATA box (*Auble and Hahn, 1993*; *Auble et al., 1994*; *Darst et al., 2001*; *Gumbs et al., 2003*; *Sprouse et al., 2006*; *Wollmann et al., 2011*; *Moyle-Heyrman et al., 2012*). A limitation of our previous work on Mot1:TBP was the finding that the crystallized state evidently represents the 'product' state after the remodeling reaction took place, but it remained unclear how Mot1 directly impacts the TBP:DNA 'substrate' state prior to the remodeling reaction (*Wollmann et al., 2011*). Obtaining a substrate state with DNA and TBP turned out to be difficult because Mot1$^{NTD}$ can disrupt TBP:DNA by its latch. However, we found that the Mot1:TBP:DNA complex is much more stable in the presence of negative cofactor 2 (NC2), another global transcriptional regulator, whose occurrence coincides with Mot1 and TBP at many genomic locations (*Andrau et al., 2002*; *Dasgupta et al., 2002*; *Hsu et al., 2008*; *Van Werven et al., 2008*; *Spedale et al., 2012*). NC2 is a heterodimer composed of α and β subunits, which highly resemble histones H2A and H2B, respectively (*Kamada et al., 2001*). We crystallized *E. cuniculi* Mot1$^{NTD}$ in complex with TBP, NC2, and a TATA promoter DNA fragment and present here the crystal structure of this complex at 3.8 Å resolution along with biochemical, electron microscopy, and cross-linking studies of the full-length Mot1 complex. Our study provides the first pseudoatomic view of a Swi2/Snf2 ATPase in complex with a DNA:protein substrate complex.

## Results

### Mot1, TBP, and NC2 form a stable complex on promoter DNA in vitro

Previous studies found that *Saccharomyces cerevisiae* Mot1 and NC2 can simultaneously bind to the TBP:DNA complex and could be isolated as a complex from yeast extracts (*Darst et al., 2003*;

*Van Werven et al., 2008*), suggesting that TBP:NC2 is a physiological substrate for Mot1. In contrast, human NC2 was reported to replace the human Mot1 homolog BTAF1 bound to TBP:DNA complexes (*Klejman et al., 2004*). These observations prompted us to explore whether *E. cuniculi* Mot1, TBP, NC2α, and NC2β form a stable complex with DNA in vitro. In line with the results demonstrated in the *S. cerevisiae* system (*Darst et al., 2003*; *Van Werven et al., 2008*), we were able to reconstitute the Mot1:TBP:NC2α:NC2β complex in the presence of a TATA box consensus sequence-containing oligonucleotide and to purify it by gel filtration (*Figure 1*). The Mot1$^{NTD}$ also formed a stable complex with TBP, NC2, and DNA (*Figure 1—figure supplement 1*). From these data, it appears that complex formation between Mot1, TBP, and NC2 on TATA DNA is evolutionary conserved. Furthermore, specific interactions between Mot1 and TBP—and not for instance interactions between the Mot1$^{CTD}$ and the DNA—are sufficient for formation of the pentameric complex.

## Mot1 dissociates the TBP:DNA:NC2 complex in the presence of ATP

Having found that NC2 stabilized the Mot1:TBP:DNA:NC2 complex, we tested if Mot1 had the ability to dissociate the TBP:DNA:NC2 complex. Consistent with the studies performed on the yeast proteins (*Darst et al., 2003*; *Van Werven et al., 2008*), *E. cuniculi* Mot1 efficiently disrupted the TBP:DNA:NC2 complex in an ATP-dependent manner (*Figure 2A*). The latch was important for this process in reactions both with and without NC2 (*Figure 2A*). Of note is that the presence of the NC2 subunits substantially increased the stability of the Mot1$^{NTD}$:TBP:DNA complex and limited Mot1$^{NTD}$'s ability to disrupt TBP:DNA with its 'latch' competitor (*Figure 2B*). However, we were not able to detect any significant changes in the Mot1-catalyzed TBP:DNA dissociation rate or the ATP hydrolysis rate of this process in the presence of NC2 (*Figure 3*). Taken together, these data show that the TBP:DNA:NC2 complex is a bona fide substrate of Mot1's remodeling activity. Moreover, in contrast to the TBP:DNA substrates, the ATPase domain is necessary to disrupt the NC2-containing complexes.

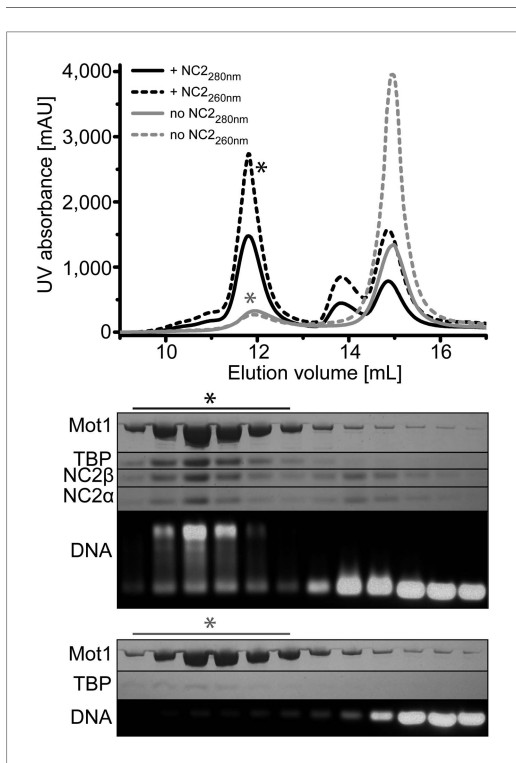

**Figure 1**. Size exclusion chromatography of the *E. cuniculi* Mot1:TBP:DNA:NC2 complex. Upper panel: elution profiles of Mot1:TBP:DNA complex (gray) and Mot1:TBP:DNA:NC2 complex (black). Absorbance at 260 nm is represented by dashed lines and at 280 nm as solid lines. Lower panel: analysis by SDS-PAGE (Coomassie staining) and agarose gel electrophoresis (Gel-Red staining). The asterisk marks fractions containing all components.

The following figure supplement is available for figure 1:

**Figure supplement 1**. Size exclusion chromatography of the *E. cuniculi* Mot1$^{NTD}$:TBP:DNA:NC2 complex.

## Organization of the Mot1$^{NTD}$:TBP: DNA:NC2 complex reveals a high level of structural conservation

Our previously reported Mot1$^{NTD}$:TBP complex structure likely represents the 'product' state of Mot1's remodeling reaction, i.e., after TBP has been dissociated from DNA (*Wollmann et al., 2011*). To capture how Mot1 binds its 'substrate' TBP:DNA complex, i.e., the interaction that is formed before the remodeling reaction takes place, we performed extensive crystallization experiments. Owing to the inherent instability of the Mot1$^{NTD}$:TBP:DNA complex, we obtained only Mot1$^{NTD}$:TBP crystals. However, the addition of NC2 efficiently stabilized the 'substrate' complex and we obtained crystals from selenomethionine-derivatized proteins diffracting to 3.8 Å resolution that contained all five components: Mot1$^{NTD}$, TBP, NC2α, NC2β, and 24 base-paired TATA box-containing DNA. The structure was solved by

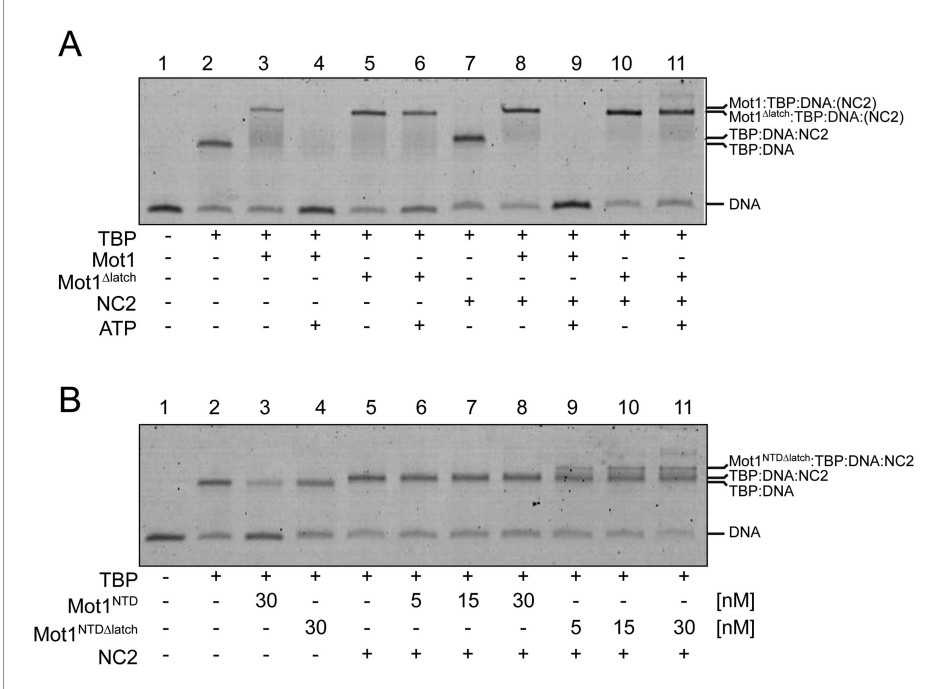

**Figure 2**. Electrophoretic mobility shift analysis of Mot1:TBP:DNA:NC2 complexes. (**A**) Upon ATP addition, Mot1 dissociated TBP from DNA (lanes 3 and 4) as well as the TBP:DNA:NC2 complex (lanes 8 and 9). Mot1$^{\Delta latch}$ was impaired in TBP removal (lanes 5 vs 6 and 10 vs 11). (**B**) Effect of NC2 on ATP-independent remodeling by Mot1$^{NTD}$. Addition of NC2 prevented the Mot1$^{NTD}$ from displacing TBP from DNA (lane 3 and 6–8). The Mot1$^{NTD\Delta latch}$ protein bound TBP:DNA more stably than Mot1$^{NTD}$ (lane 4). Addition of NC2 to Mot1$^{NTD\Delta latch}$:TBP: DNA complex resulted in a distinct shift (lanes 9–11), consistent with formation of a stable complex containing all components.

molecular replacement using *E. cuniculi* Mot1$^{NTD}$ and TBP (*Wollmann et al., 2011*) as the search models. The initial fragments of DNA and NC2—based on the human TBP:DNA:NC2 (*Kamada et al., 2001*) and NF-Y complex (*Nardini et al., 2013*)—were manually fitted according to the difference density, refined as rigid bodies, and extended. We used the feature-enhanced $2F_o - F_c$ maps implemented in *PHENIX* to reduce the model bias and to enable unambigous density interpretation (*Afonine et al., 2015*). The final model was refined to $R_{work}/R_{free}$ of 23.5/25.8% with good stereochemistry (*Table 1*) and includes 39 out of the 48 DNA bases as well as 89% of all protein residues. The sequence register was confirmed by computing anomalous difference density map that showed signal of the selenium atoms (*Figure 4—figure supplement 1B*).

In the crystal structure, we find one copy each of Mot1$^{NTD}$, TBP, NC2α, NC2β, and 24 base pairs of DNA, in a semi-compact complex with approximate dimensions of 100 Å × 95 Å x 95 Å (*Figure 4*). Mot1 and NC2 occupy opposite binding surfaces on TBP and completely encircle the DNA-bound TBP molecule, which recognizes the minor groove of the TATA sequence. The Mot1$^{NTD}$ is oriented towards the upstream DNA (the 5′ end of the TATA box-containing strand) and predominantly binds the convex ('top') side of TBP. Furthermore, like in the human TBP:DNA: NC2 complex, NC2 locks TBP onto the promoter utilizing its histone fold domain (NC2$^{HF}$), which is bound to the underside of the DNA, and the C-terminal helix *H5* of NC2β, which reaches around the DNA and binds the convex side of TBP (*Kamada et al., 2001*). Of note is that no interpretable electron density of the Mot1 latch residues 98–142 is observed. Therefore, it is probable that the latch is ordered only after binding to TBP's DNA-binding surface. Taken together, the *E. cuniculi* Mot1$^{NTD}$:TBP:DNA:NC2 crystal structure shows a high degree of evolutionary conservation of the TBP:NC2 interaction on a structural level. Moreover, the encircling of TBP: DNA by NC2 explains the resistance of Mot1$^{NTD}$:TBP:DNA:NC2 complex to premature dissociation caused by the latch.

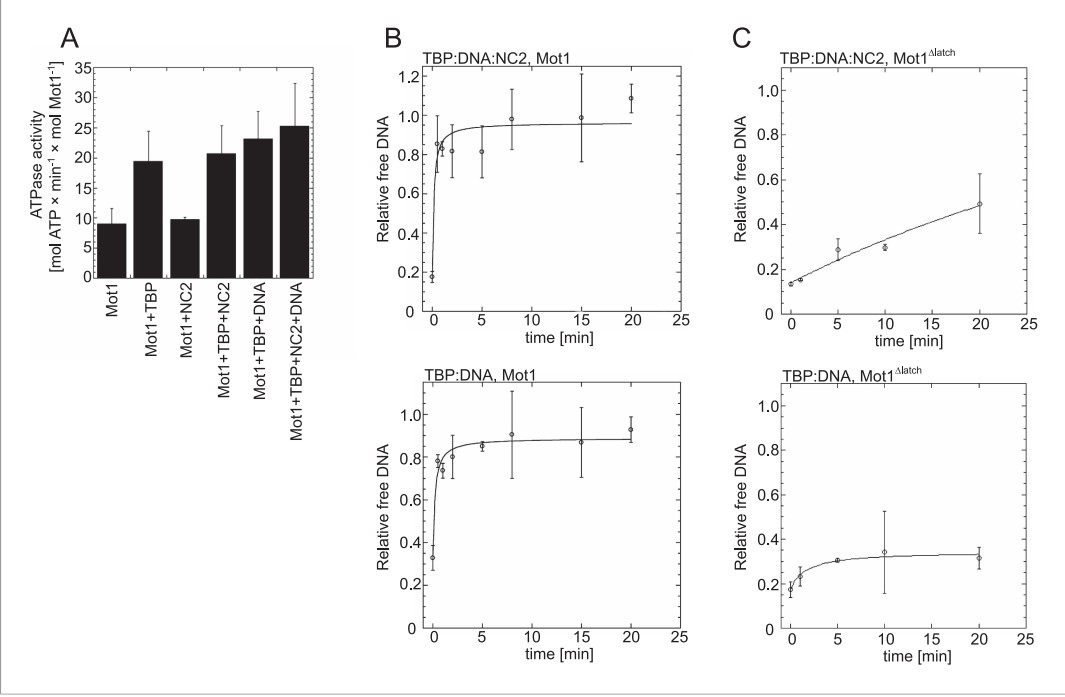

**Figure 3**. Effect of NC2, TBP, and DNA on Mot1's activity. (**A**) The graph shows the steady-state ATPase activity of 8 nM Mot1 alone or in the presence of 80 nM NC2 and/or TBP and with or without 23 nM TATA-containing DNA. The assays were performed as described previously (**Wollmann et al., 2011**). The data represent the mean ± the standard deviation obtained from at least two independent experiments. (**B**) The graphs show the rates of Mot1-catalyzed TBP:DNA:NC2 dissociation (top) and TBP:DNA dissociation (bottom) measured by quantitation of the free DNA level by EMSA following ATP addition to pre-formed complexes. The assays were performed as previously described (**Wollmann et al., 2011**). (**C**) The same as in (**B**) shown for the Mot1$^{\Delta latch}$ mutant.

## Mot1 binding induces changes in the interaction interfaces in the 'substrate' complex

The Mot1-bound 'substrate' state superficially appears to be a superposition of the Mot1$^{NTD}$:TBP 'product' state and the TBP:DNA:NC2 'substrate' complex, with the important exception that the latch in the DNA-bound complex is displaced (*Figure 5A*). Although the contacts of Mot1$^{NTD}$ and NC2 within the complex are mainly TBP- and/or DNA-mediated, which is consistent with the lack of a stable interaction between Mot1 and NC2 in the absence of DNA in vitro (data not shown), we observed several notable conformational differences. In comparison with the Mot1$^{NTD}$:TBP 'product' state, the interaction between the Mot1$^{NTD}$ and the convex site of TBP prominently extends towards the C-terminus of Mot1$^{NTD}$, engaging HEAT repeats 4 to 10 and the insertion domain. Additionally, the 'anchor' helix *H5* of NC2β bound to the upper side of TBP contacts the insertion domain of Mot1$^{NTD}$, the only non-HEAT repeat stretch in the N-terminal part of Mot1 (*Figure 5B* and *Figure 5—figure supplement 1A*). As a result, the interaction interface between Mot1$^{NTD}$ and TBP's convex surface increases dramatically from ~900 Å$^2$ to ~1500 Å$^2$, as calculated using the PISA server (*Krissinel and Henrick, 2007*), thus suggesting that Mot1 might even have a tighter grip on TBP in the presence of NC2. The additional interactions between the loops and structural elements of the Mot1$^{NTD}$ and TBP result in a considerable compaction of the α-helical HEAT repeat array. Therefore, since the capability to rearrange was shown to be an essential property of long α-helical solenoids, shape adaptation and conformational changes of the Mot1's HEAT domain might play an important role in the dissociation mechanism (*Grünwald et al., 2013*).

Interestingly, the kink of the longitudinal axis of the DNA duplex introduced by TBP appears to be less severe than the ~90° bend observed in other TBP:DNA complexes and is slightly underwound (*Figure 5C* and *Figure 5—figure supplement 1B*) bringing the upstream DNA closer to the N- and

**Table 1.** Data collection and refinement statistics

| Data collection* | |
|---|---|
| Space group | C 1 2 1 |
| Cell dimensions | |
| $a$, $b$, $c$ (Å) | 150.6, 140.3, 90.8 |
| $\alpha$, $\beta$, $\gamma$ (°) | 90.0, 113.7, 90.0 |
| Resolution (Å) | 49.2–3.8 (4.0–3.8)† |
| $R_{merge}$ (%) | 10.4 (78.9) |
| $CC_{(1/2)}$ | 99.8 (83.8) |
| $I/\sigma I$ | 7.5 (1.5) |
| Completeness (%) | 98.2 (93.5) |
| Redundancy | 3.4 (3.4) |
| Refinement | |
| Resolution (Å) | 49.2–3.0 (4.0–3.8) |
| No. reflections | 17,163 |
| $R_{work}$/$R_{free}$ (%) | 23.5/25.8 (26.9/30.2) |
| No. atoms | |
| Protein | 8422 |
| DNA | 799 |
| Ligand/ion | 0 |
| Water | 0 |
| Isotropic $B$-factors (Å$^2$) | |
| Protein | 69 |
| DNA | 135 |
| R.m.s. deviations | |
| Bond lengths (Å) | 0.009 |
| Bond angles (°) | 0.75 |

*From one crystal.

†Values in parentheses are for the highest-resolution shell.

C-termini of Mot1$^{NTD}$ (the 'gap' of the horseshoe, *Figure 5—figure supplement 1C*). This observation is of potential importance since the upstream part of DNA binds to the Mot1's Swi2/Snf2 domain, which immediately follows the C-terminus of Mot1$^{NTD}$ (*Auble and Hahn, 1993*; *Auble et al., 1994*; *Sprouse et al., 2006*; *Darst et al., 2001*; *Wollmann et al., 2011*). Furthermore, the linker helix *H4* of NC2β adopts a partially unfolded conformation and its interaction with the major groove of the downstream DNA is lost (*Figure 5D*). Thus, Mot1 might influence the C-terminal stirrup of TBP indirectly via interaction with NC2. (*Figure 5—figure supplement 1B*). Moreover, due to the changed DNA geometry, NC2$^{HF}$ is moved and rotated with respect to TBP. Despite this change, the direct interaction between the NC2$^{HF}$ and DNA does not seem to be affected and is similar to the interaction of histones H2A/H2B with DNA in the nucleosome (*Luger et al., 1997*). However, it has to be noted that both sides of the DNA molecule are engaged in the formation of crystal contacts; thereby, we cannot exclude the possibility that the DNA conformation is affected by crystal packing. Nevertheless, it is unlikely that crystal lattice formation can entirely explain the change in DNA trajectory (see 'Discussion'). In summary, in the Mot1$^{NTD}$:TBP:DNA:NC2 complex crystal structure, we observe a variety of small to medium conformational changes. The observed changes indicate reduced strength of the TBP–DNA and DNA–NC2 interactions as well as stabilization of Mot1–TBP interface due to Mot1$^{NTD}$ binding.

## In the presence of ATP analogs Mot1 ATPase domain adopts a 'closed' conformation

Upstream DNA was shown to play a crucial role not only in the stabilization of Mot1:TBP:DNA complexes, but also in the ATP-mediated dissociation in vitro (*Auble and Hahn, 1993*; *Darst et al., 2001*; *Gumbs et al., 2003*; *Sprouse et al., 2006*; *Moyle-Heyrman et al., 2012*). Additionally, it is known that the nucleotide state of Swi2/Snf2 domains modulates their conformation and affinity for DNA (*Lewis et al., 2008*; *Moyle-Heyrman et al., 2012*). Therefore, in order to define the location of the Swi2/Snf2 domain in the Mot1:TBP:DNA:NC2 complex, we analyzed the full-length complex by chemical cross-linking combined with mass spectrometry (CX-MS) as well as by negative stain electron microscopy (EM, see below).

To facilitate this analysis, we formed the Mot1:TBP:DNA:NC2 complex on long TATA box-containing DNA (at least 26 base pairs upstream from the TATA box) and used ADP·BeF$_X$, ATPγS, and ADP to lock the ATPase domain in ATP- and ADP-bound states (*Ponomarev et al., 1995*). In our analysis, we identified 133, 129, and 97 inter- and intra-protein cross-links within the ADP·BeF$_X$-, ATPγS- and ADP-bound Mot1:TBP:DNA:NC2 complexes, originating from 116, 109, and 82 non-redundant lysine linkage pairs, respectively (*Figure 6—figure supplement 1*, *Table 2* and *Table 2—source data 1*). Most of the cross-links could be placed within the crystal structure or within the Mot1$^{CTD}$. The cross-links that were detected between the latch (Lys115 or Lys138) and other components could not be mapped since the latch is disordered in the crystal structure. The latch

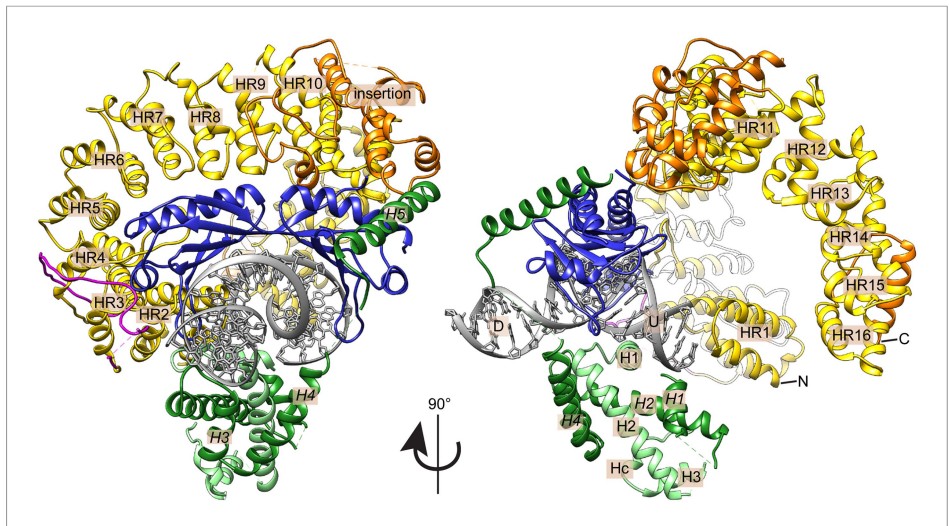

**Figure 4**. Crystal structure of the *E. cuniculi* Mot1^NTD:TBP:DNA:NC2 complex. Front and side views of the structure, represented as cartoon models with highlighted secondary structure. TBP (blue), NC2α (light green), and NC2β (dark green) encircle kinked promoter DNA (gray with upstream (U) and downstream (D) regions labeled). The HEAT repeats (HR, yellow with N- and C-termini marked) and insertion domain (orange) of the Mot1^NTD bind along the convex surface of TBP and contact the C-terminal helix (*H5*) of NC2β. The latch of Mot1^NTD (magenta), which has been previously shown to bind to TBP's DNA-binding cleft, is mostly disordered in the presence of DNA and NC2.

The following figure supplement is available for figure 4:

**Figure supplement 1**. X-ray electron density maps.

formed contacts with TBP, Mot1^CTD as well as NC2^HF, consistent with its high flexibility even in the context of full-length Mot1. The remaining cross-links were detected between Mot1^CTD and the rest of the complex. As an example, in all data sets, we found numerous cross-links between the Mot1^CTD and NC2^HF, mainly between the helix Hc and loop L2 (joining helices H3 and Hc) of NC2α and RecA2 subdomain of Mot1^CTD (*Figure 6A*).

As shown in *Figure 6B*, in the crystal structures of Swi2/Snf2 ATPase domains, the two RecA-fold subdomains were observed to adopt a variety of different positions with respect to each other, ranging from 'open' and 'semi-closed' to 'closed' (*Dürr et al., 2005*; *Thomä et al., 2005*; *Shaw et al., 2008*; *Hauk et al., 2010*). Importantly, on the basis of related SF2 helicases, ATP was shown to bind to the interface of the RecA-fold subdomains and stabilize a 'closed' state (*Sengoku et al., 2006*). The CX-MS approach can provide insights into the architecture of protein complexes with domain to motif resolution (*Nguyen et al., 2013*; *Tosi et al., 2013*; *Politis et al., 2014*). Therefore, to gain insights into the conformation of the Swi2/Snf2 domain in different nucleotide states, we analyzed the 32 cross-links that we detected between the two RecA-fold subdomains within Mot1^CTD and mapped them onto homology models of Mot1^CTD in different conformations, based on 'open', 'semi-closed' and 'closed' states. Among these cross-links, 18 do not distinguish between these conformations, since they are all either below (14) or above (4) the distance cutoff of 30 Å, which accounts for the length of the cross-linker and two lysine side chains (*Politis et al., 2014*, *Table 3* part A). From the cross-links, which do distinguish between the conformations, two were present in all three data sets (ADP, ATPγS, ADP·BeF_x) and are thus non-informative (*Table 3* part B). However, the remaining 4 cross-link sites (8 cross-links total) were detected only in the presence of ADP·BeF_x and ATPγS (*Table 3* part C). Notably, all of these cross-links, which were absent from the ADP data set, are only consistent with the 'closed' conformation (*Figure 6C*, *Figure 6—figure supplement 2*). Thus, our data suggest that the ATP-mimicking ADP·BeF_x and ATPγS analogs stabilize a more 'closed' conformation. Since we did not detect any cross-links that would be unique for the ADP-supplied sample, whether Mot1^CTD adopts a distinct state in the presence of this nucleotide or is simply more flexible remains to be addressed in future studies.

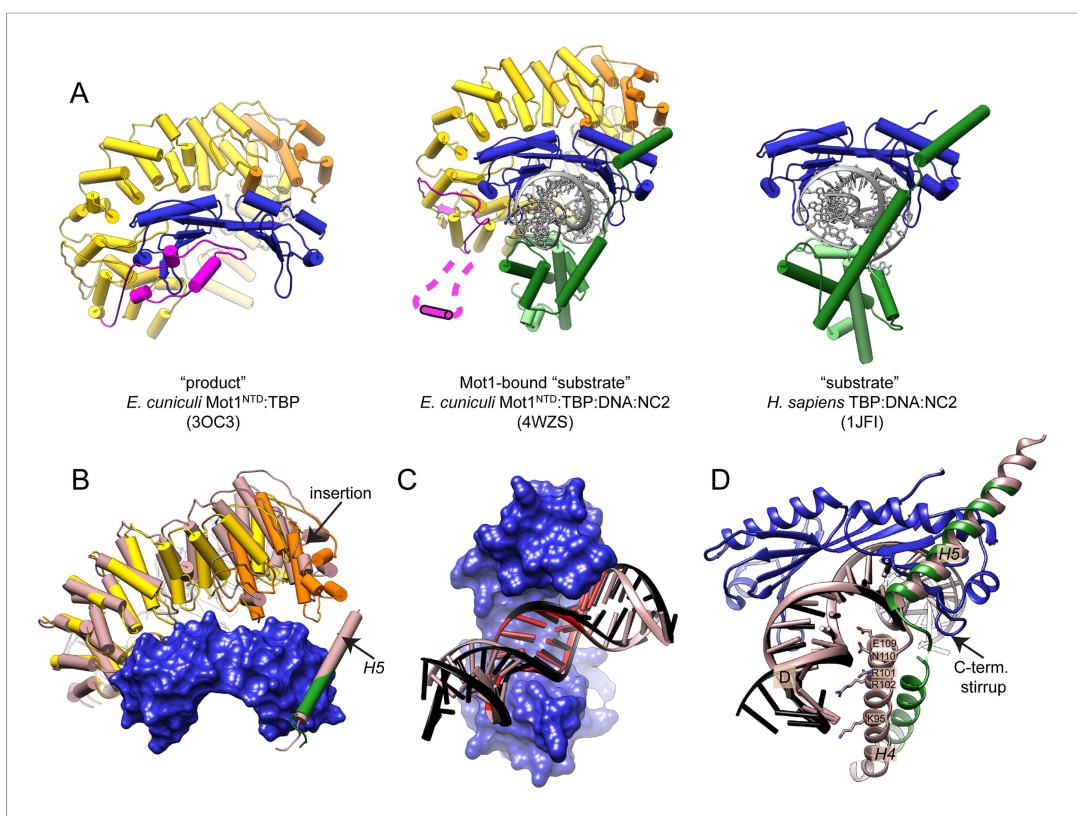

**Figure 5**. Features of the Mot1-bound 'substrate' complex. (**A**) Comparison of the 'product' (left, *Wollmann et al., 2011*), 'substrate' (right, *Kamada et al., 2001*), and Mot1-bound 'substrate' complex (center). Mot1's latch in the latter structure is disordered and represented schematically. Color code is as in *Figure 4*. Panel (**B**) shows Mot1NTD and TBP from the Mot1NTD:TBP:DNA:NC2 structure (color code as in *Figure 4*) superimposed via TBP with the 'product' Mot1NTD:TBP structure and NC2β *H5* from the 'substrate' TBP:DNA:NC2 structure (both shown in light brown). Mot1NTD shifts toward *H5* and TBP although the position of *H5* is not affected. Panels (**C**) and (**D**) show conformational changes of NC2 and DNA in the Mot1NTD:TBP:DNA:NC2 complex structure (color code as in *Figure 4*) compared to the TBP:DNA:NC2 'substrate' complex (shown in light brown) superimposed via TBP. (**C**) View from the concave side of TBP shows that in the presence of Mot1, DNA is partially straightened and underwound. TATA box region is highlighted in red. Panel (**D**) shows that partially unfolded helix *H4* of NC2β, which joins the NC2HF with helix *H5*, loses its interaction with downstream DNA and is close to TBP's C-terminal stirrup.

The following figure supplement is available for figure 5:

**Figure supplement 1**. Mot1NTD:TBP:DNA:NC2 structure features.

## EM and CX-MS provide important insights into Mot1CTD localization and orientation

We next visualized the Mot1:TBP:DNA:NC2 complex by electron microscopy. We calculated a negative stain reconstruction of the Mot1:TBP:DNA:NC2 complex in the presence of ADP·BeFX and experimentally determined its handedness (*Figure 7—figure supplement 1*). The 22 Å reconstruction, of overall dimensions of 115 Å × 115 Å × 100 Å, is in good agreement with the size of the partial complex observed in the crystal structure. Due to better resolution, the characteristic 'C'-shaped structure of Mot1NTD is more pronounced than in the reconstruction of the *E. cuniculi* Mot1:TBP and *H. sapiens* BTAF1:TBP complexes (*Pereira et al., 2004*; *Wollmann et al., 2011*). Furthermore, to place the Mot1NTD:TBP:DNA:NC2 crystal structure model into the EM density, we employed an unsupervised rigid body docking approach using the *Situs* software package (*Wriggers, 2010*), which resulted in a convincing solution (*Figure 7A*). The prominent and centered additional density, which is localized in the immediate vicinity of the N- and C-terminal ends of Mot1NTD

**Table 2**. Localization of the cross-links identified in Mot1:TBP:DNA:NC2 complex

| Experiment | Crystal structure | Within Mot1CTD | | Latch-crystal structure | Latch– Mot1CTD | Crystal structure– Mot1CTD | Total | Decoy§ | Estimated FDR [%] |
| | | Intralobe* | Interlobe† | | | | | | |
|---|---|---|---|---|---|---|---|---|---|
| ADP·BeFx | 46 (42) | 21 (17) | 15 (12) | 8 (8) | 5 (5) | 37 (31) + 1 (1)‡ | 133 (116) | 2 | 0.8 |
| ATPγS | 51 (44) | 17 (14) | 10 (9) | 11 (11) | 3 (3) | 37 (28) | 129 (109) | 1 | 1.5 |
| ADP | 40 (36) | 14 (11) | 7 (7) | 11 (10) | 4 (3) | 21 (15) | 97 (82) | 2 | 2.0 |

*Within RecA1 or RecA2 subdomain.
†Between RecA1 and RecA2 subdomain.
‡Between Mot1CTD and the linker joining N- and C-terminal domains (could not be mapped).
§Detected from a reverse database, estimating false-discovery rate.
Numbers refer to the total number of cross links, including cross-linked sites which were detected more than once (i.e., from miss-cleaved peptides).
Numbers in brackets refer to non-redundant linkages only.

**Source data 1**. Full list of the detected cross-links.

(the 'gap' of the horseshoe), likely harbors the C-terminal Swi2/Snf2 domain (*Figure 7B*). To assess the placement of Mot1CTD, we again used our CX-MS data. Using *RANCH* (*Petoukhov et al., 2012*), we first generated a set of 20,000 theoretical Mot1:TBP:DNA:NC2 assemblies allowing flexible movement of the eight amino acid linker joining the Mot1CTD model and Mot1NTD. As suggested by the interpretation of the CX-MS data, for this analysis we used the Mot1CTD modeled in the 'closed' conformation. On each of the computed models, we next mapped all of the non-redundant cross-links between Mot1CTD and the rest of the complex. We repeated this procedure for the ATPγS and ADP·BeFx data sets independently (using 28 and 31 cross-links, respectively). We then performed violation scoring by applying the 30 Å cutoff distance. Convincingly, for both of the independently analyzed data sets exactly the same models were among best-scored ensembles (the lowest violation scores, *Figure 7—figure supplement 2*).

The EM- and CX-MS-derived placements not only converged, but are also in excellent agreement with the localization that has been previously determined in biochemical assays (*Auble and Hahn, 1993*; *Auble et al., 1994*; *Darst et al., 2001*; *Gumbs et al., 2003*; *Sprouse et al., 2006*; *Wollmann et al., 2011*; *Moyle-Heyrman et al., 2012*). In such an arrangement, the upstream DNA engaged by Mot1CTD directly continues from the upstream DNA of the Mot1NTD:TBP:DNA:NC2 complex (*Figure 7C,D*). Accounting for slight rearrangements within the HEAT repeats, the analysis implies that Mot1CTD would contact DNA not further than ~20 bp upstream from the TATA sequence. This would be in agreement with the requirement reported for the yeast Mot1, which was shown to contact around 17 bp upstream from the TATA box (*Wollmann et al., 2011*). It is, however, not clear if the upstream NC2-bound DNA adopts B-DNA form or, like in case of a related histone fold transcriptional factor NF-Y, has bent, nucleosome-like curvature (*Huber et al., 2012*; *Nardini et al., 2013*).

## Mot1 locally dissociates the TBP:DNA:NC2 complex

Swi2/Snf2 ATPase motors couple ATP hydrolysis to translocation along dsDNA (*Saha et al., 2002*; *Whitehouse et al., 2003*; *Zofall et al., 2006*). The release of TBP from DNA by Mot1, however, does not appear to involve highly processive ATP-dependent DNA tracking (*Auble and Steggerda, 1999*). The changes in the TBP:DNA and NC2:DNA interactions observed in the Mot1NTD:TBP:DNA:NC2 crystal structure suggest that Mot1NTD binding leads to an alteration of the interaction with promoter DNA. Such a potential destabilization—due to the ring that is formed around DNA by TBP and NC2—does not generally result in removal from TBP from DNA, consistent with our biochemical observations. However, it is possible that TBP:NC2 has an increased lateral mobility on DNA. Interestingly, in single-molecule experiments, NC2 induces dynamic conformational changes in the TBP:DNA interface, enabling TBP to laterally move on DNA substrates (*Schluesche et al., 2007*). Consequently, NC2-induced lateral mobility of TBP was proposed to explain how NC2, by sliding TBP, could lead to TBP repositioning from/towards the promoter sites. Although the underlying conformational changes have not been seen in the TBP:DNA:NC2 crystal structure, our structural

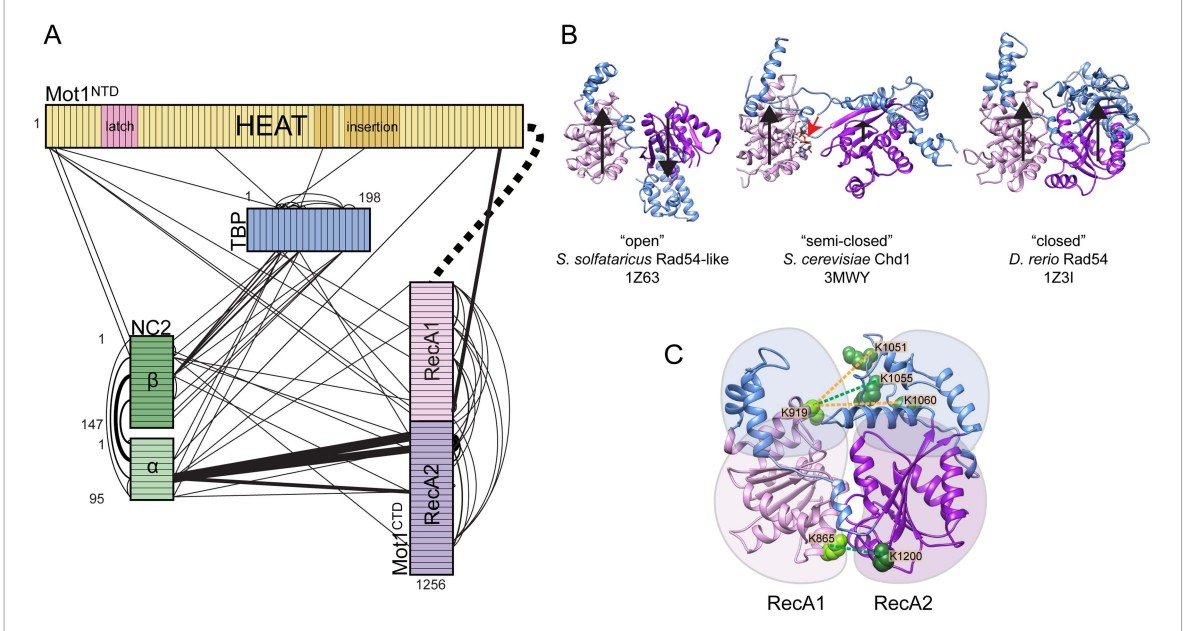

**Figure 6**. Mot1:TBP:DNA:NC2 complex analyzed by CX-MS. **(A)** General topology of the Mot1:TBP:DNA:NC2 complex in the presence of ATPγS derived from the CX-MS data. Each polypeptide is divided into 10 amino acid segments. The black solid lines represent the identified cross-links. For simplicity, cross-links to the latch region of Mot1 are not displayed. Line thickness is proportional to the number of cross-links detected between joined segments. The dashed line between Mot1<sup>NTD</sup> and Mot1<sup>CTD</sup> represents the eight amino acid linker between these domains. **(B)** Crystal structures of Swi2/Snf2 domains used for the cross-link analysis: *Sulfolobus solfataricus* Rad54-like in an 'open' conformation (*Dürr et al., 2005*), *Saccharomyces cerevisiae* Chd1 in a 'semi-closed' conformation (*Hauk et al., 2010*) in the presence of ATPγS (red arrow) and *Danio rerio* Rad54 in a 'closed' conformation (*Thomä et al., 2005*). PDB accession codes are included. The structures are oriented with respect to the RecA1 subdomain. Relative orientations of the RecA1 (pink) and RecA2 (purple) subdomains are represented by the black arrows. Family-specific insertion regions are indicated in blue. Auxiliary domains were omitted. Panel **(C)** shows cross-linked sites listed in *Table 3* part C mapped on the Swi2/Snf2 domain modeled in the 'closed' conformation. Cross-links detected only in the presence of ADP·BeF$_x$ and ATPγS are shown as green dashed lines, whereas the cross-links present only in the ADP·BeF$_x$ data set are in orange.

The following figure supplements are available for figure 6:

**Figure supplement 1**. Titration of the cross-linking agent disulfosuccinimidyl glutarate (DSSG).

**Figure supplement 2**. Analysis of the cross-links between RecA1 and RecA2 subdomains within the Mot1<sup>CTD</sup> models.

analysis indeed indicates changes in the DNA trajectory that are consistent with increased lateral mobility. Thus, the analysis of our structural data suggested two models for the observed dissociation of the Mot1:TBP:DNA:NC2 complex in the presence of ATP. One possibility is that Mot1 uses a Swi2/Snf2 translocase activity to translocate the TBP:NC2 ring along DNA like a sliding clamp without disrupting it. In our EMSAs, the complex would eventually dissociate from DNA when the DNA end is reached. Alternatively, Mot1-induced conformational changes in the TBP:DNA:NC2 substrate could directly dissociate the complex from DNA without translocation along the DNA. To distinguish between these two options, we performed EMSAs using digoxygenin-labeled DNA, the ends of which were blocked with an anti-digoxygenin antibody. The DNA probes used were not much longer than the minimal DNA required for the formation of the complex, consisting of just 30 bp of DNA upstream of the TATA box and 9 bp of DNA downstream. As shown in *Figure 8*, we observed no significant impact of the presence or placement of antibody block on either side of the DNA on the efficiency of the remodeling reaction. Therefore, in this setting Mot1 does not appear to apply highly processive sliding of TBP:NC2 along DNA but rather utilizes a more direct, local disruption mechanism consistent with previous studies of TBP–DNA complexes (*Auble and Steggerda, 1999*). Since NC2 and TBP encircle the DNA and do not interact in the absence of DNA, Mot1 likely disrupts the TBP: NC2 interaction as well.

**Table 3**. Distances of the cross-links detected between RecA1 and RecA2 subdomains of Mot1$^{CTD}$ mapped on different structural models

| | Detected linkages | | Euclidean C$_\alpha$–C$_\alpha$ distance [Å] | | | Total number of detected cross links | | |
|---|---|---|---|---|---|---|---|---|
| | Residue 1 | Residue 2 | SsoRad54-like ('open') | DrRad54 ('closed') | ScChd1 ('semi-closed') | ADP·BeF$_x$ | ATPγS | ADP |
| A | 796 | 1013 | 16 | 22 | 20 | 1 | 1 | 1 |
| | 796 | 1200 | 25 | 17 | 16 | 1 | 1 | 0 |
| | 842 | 1055 | 58 | 52 | 52 | 0 | 1 | 0 |
| | 864 | 1039 | 39 | 49 | 50 | 1 | 1 | 1 |
| | 1003 | 1013 | 12 | 11 | 10 | 1 | 0 | 1 |
| | 1003 | 1200 | 15 | 18 | 16 | 2 | 1 | 1 |
| | 1008 | 1200 | 12 | 14 | 12 | 2 | 0 | 1 |
| B | 864 | 1200 | 34 | 24 | 24 | 1 | 1 | 1 |
| | 919 | 1086 | 25 | 42 | 47 | 1 | 1 | 1 |
| C | 865 | 1200 | 31 | 20 | 21 | 1 | 1 | 0 |
| | 919 | 1051 | 63 | 22 | 43 | 1 | 0 | 0 |
| | 919 | 1055 | 57 | 19 | 36 | 2 | 2 | 0 |
| | 919 | 1060 | 51 | 25 | 36 | 1 | 0 | 0 |

Part A shows the cross links, which do not distinguish between the conformations. Part B and C list cross-links, for which the mapped distances were significantly different depending on the model used (i.e., <30 Å for one model and >30 Å for another). Eight of these cross-links (shown also in **Figure 6C**) were detected only in the presence of ADP·BeF$_x$ and ATPγS and are listed in part C.

## Discussion

Swi2/Snf2-type ATPases remodel protein:DNA complexes to regulate the structure and epigenetic state of chromatin regions during transcription, DNA replication, and DNA repair. Mot1, apart from being an essential gene regulator, serves as an attractive model system for structural and mechanistic studies of Swi2/Snf2 enzymes. Here, we provide a pseudoatomic structure of the Swi2/Snf2 enzyme Mot1 bound to its protein:DNA substrate and a detailed structural framework for the interaction of Mot1 with the TBP:DNA:NC2 complex.

Swi2/Snf2 enzymes typically remodel protein:DNA complexes by disrupting the interface between the nucleic and protein partners of the target complex. ATP hydrolysis-dependent translocation of the Swi2/Snf2 ATPase domain could provide most if not all of the chemomechanical force, but a question is whether binding of the remodeler induces conformational changes in the target. Indeed, we find that the association of Mot1 with the TBP:DNA:NC2 complex results in some unexpected conformational changes that may have implications for understanding the ATP-dependent dissociation mechanism. Binding of Mot1 leads to subtle but consistent conformational changes between substrate proteins and DNA, which could prime the complex for ATP-dependent disruption. Although the interface between Mot1 and NC2 that we observe in the crystal structure of the Mot1$^{NTD}$:TBP:DNA:NC2 complex is arguably very small, the contact between Mot1's insertion domain and NC2 could help separate NC2's main TBP anchor helix *H5* from TBP and *H4* from DNA. In support of this, the very C-terminal unstructured region of NC2β following helix *H5* has been previously shown to be responsible for the repressive role of NC2 (*Yeung et al., 1994*; *Yeung et al., 1997*).

Additionally, compared to the state in the absence of Mot1, the conformation of the upstream DNA appears to be altered. Indeed, recent experiments have shown that Mot1:TBP:DNA and TBP:DNA:NC2 complexes are mobile and flexible due to an equilibrium between bent and unbent DNA states (*Schluesche et al., 2007*; *Moyle-Heyrman et al., 2012*). However, consistent with the observation that Mot1-induced dynamic DNA behavior in the TBP:DNA complex occurs also on DNA templates which are too short to directly contact the ATPase domain (*Moyle-Heyrman et al., 2012*), we propose that even minimal Mot1-dependent conformational changes could have a critical impact on the TBP–TATA box interaction, especially as the severely bent TATA box itself might accelerate

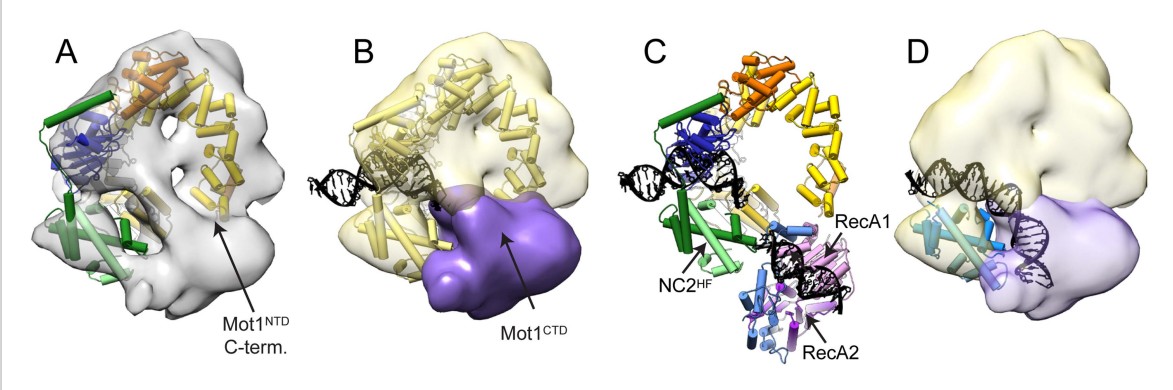

**Figure 7**. Pseudoatomic model for the Mot1:TBP:DNA:NC2 complex. Panels (**A**) and (**B**) show the Mot1NTD:TBP:DNA:NC2 crystal structure rigid body docked into the negative stain EM map of the Mot1:TBP:DNA:NC2:ADP·BeFx complex. (**B**) The density segment corresponding to Mot1CTD (purple) is in a direct proximity to the C-terminal end of Mot1NTD and to NC2HF. The transparent yellow segment corresponds to the Mot1NTD:TBP:NC2 module. The TBP-bound promoter DNA fragment from the crystal structure is included (black). (**C**) Orientation of Mot1CTD derived from the interpretation of the CX-MS data. For simplicity, one of the best-scoring models is shown. RecA1 (pink) and RecA2 (purple) correspond to the subdomains of a Swi2/Snf2 fold. Protrusions are indicated in blue. The DNA fragment bound to Mot1CTD is modeled based on the *Sso*Rad54-like:DNA crystal structure by superimposing via the RecA1 subdomain (*Dürr et al., 2005*). (**D**) Overlay of the EM map segments from (**B**) and the crystal structure of NF-YB/NF-YC transcription factor (blue) bound to a DNA fragment (*Nardini et al., 2013*) superimposed via the histone fold of NC2. Color coding for Mot1NTD, TBP, and NC2 shown in (**A**) and (**C**) is the same as in *Figure 4*.

The following figure supplements are available for figure 7:

**Figure supplement 1**. EM data.

**Figure supplement 2**. Analysis of Mot1CTD localization based on the CX-MS data.

dissociation by acting as a 'spring' for rapid release of TBP (*Tora and Timmers, 2010*). This would be in line with, for example, DNase I footprinting experiments, which showed that Mot1 binding alters TBP's TATA DNA protection pattern (*Auble and Hahn, 1993*; *Darst et al., 2001*; *Sprouse et al., 2006*), its ability to discriminate between classical and mutated TATA sequences, and why Mot1:TBP: DNA complexes can be formed by TBP mutants which are defective for TBP:DNA and Mot1:TBP complex formation (*Gumbs et al., 2003*; *Klejman et al., 2005*).

We also note that the conformation of TBP bound to DNA in the presence of Mot1 and NC2 is somewhat distinct from that of TBP bound to DNA alone and more similar to TBP bound to Mot1 alone ('product' complex, *Figure 5—figure supplement 1D*), although TBP remains in a rather similar conformation. The idea that Mot1 might directly influence TBP structure has been broadly discussed, although there has not yet been any direct evidence for it (*Auble et al., 1997*; *Adamkewicz et al., 2000*; *Darst et al., 2003*; *Gumbs et al., 2003*; *Sprouse et al., 2006*; *Auble, 2009*). Therefore, inducing substantial conformational changes in the conserved TBP core domain seems to be rather unlikely. Nevertheless, according to our observations, Mot1 could still indirectly perturb TBP's C-terminal stirrup, thereby affecting phenylalanine residues responsible for sharp DNA bending (*Kim et al., 1993*).

NC2 prevents assembly of TBP with other general transcription factors (*Meisterernst and Roeder, 1991*; *Inostroza et al., 1992*; *Cang et al., 1999*; *Kim et al., 2008*) but is retained on DNA in the absence of functional Mot1 in vitro and in vivo (*Geisberg et al., 2002*; *Schluesche et al., 2007*; *Van Werven et al., 2008*; *de Graaf et al., 2010*). While Mot1 forms a stable complex with DNA-bound TBP and NC2, it is fully capable of disrupting this complex in the presence of ATP. Thus, our data suggest that the TBP:DNA:NC2 complex is not only a *bona fide* but perhaps even preferential substrate for Mot1. It is possible that NC2 marks the TBP-containing complexes for Mot1-catalyzed disassembly, e.g., to prevent the formation of aberrant transcription preinitiation complexes on intragenic regions (*Van Werven et al., 2008*; *Spedale et al., 2012*; *Koster et al., 2014*; *Koster and Timmers, 2015*). In this context, it is important to note that NC2, similarly to histones, has been shown to undergo phosphorylation, acetylation, methylation, and ubiquitination in vivo (*Dephoure et al., 2008*;

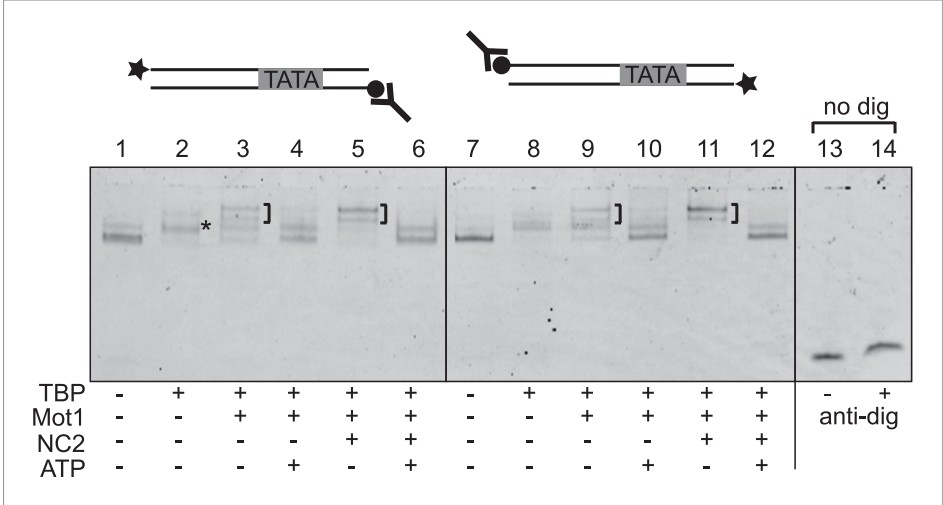

**Figure 8**. Mot1-mediated displacement of TBP and NC2 from end-blocked DNA templates. Electrophoretic mobility shift analysis of ATP-dependent disruption of Mot1:TBP:DNA complexes with and without NC2 as in *Figure 2*, but using DNA substrates that carry a digoxygenin label at one end (circle) and fluorescein for DNA detection at the other end (star). Reactions in lanes 13 and 14 were performed using DNA alone without digoxygenin modification, and demonstrate that the addition of digoxygenin antibody (Y-shape) had no effect on the mobility of unmodified DNA. Blocking of either end by the antibody did not result in a detectable decrease in TBP dissociation activity, suggesting that Mot1 does not translocate an intact TBP:NC2 clamp along DNA but locally disrupts the TBP:DNA:NC2 complex.

*Wang et al., 2008*; *Van Hoof et al., 2009*; *Huttlin et al., 2010*; *Kim et al., 2011*; *Alcolea et al., 2012*; *Wagner et al., 2012*; *Zhou et al., 2013*; *Guo et al., 2014*; *Sharma et al., 2014*). Therefore, the impact of NC2 modifications on Mot1 recruitment and activity needs to be further investigated, especially given that the remodeling activity of Swi2/Snf2 family members can be controlled by post-translational modification of histone substrates or remodeler subunits themselves (*Clapier et al., 2002*; *Ferreira et al., 2007*; *Dutta et al., 2014*).

Importantly, the CX-MS analysis uncovered unanticipated interactions between Mot1[CTD] and NC2. The position of Mot1[CTD] relative to NC2[HF] in our CX-MS- and EM-derived models is analogous to the association of a Swi2/Snf2 domain bound to superhelical location 2 (SHL2), i.e., in direct proximity to the H3/H4 histone pair slightly away from the dyad axis (*Dang and Bartholomew, 2007*; *Dechassa et al., 2012*). Notably, most of the cross-links between the Mot1[CTD] and NC2[HF] map to the protrusion region of the RecA2-like subdomain, the major family-specific insertion region of Swi2/Snf2 ATPases (*Flaus et al., 2006*). A general role for the Swi2/Snf2 protrusions in distorting local DNA structure has been proposed (*Hauk and Bowman, 2011*). For example, Snf2 has been suggested to disturb histone:DNA contacts by wedging the RecA2 lobe between DNA and protein substrates (*Dechassa et al., 2012*). In our model, the Swi2/Snf2 protrusions, especially from the RecA2 subdomain, are well positioned to directly affect DNA and NC2:DNA contacts. Therefore, the close proximity of the Mot1 Swi2/Snf2 domain to the NC2 histone fold reveals interesting parallels to the proposed interactions of nucleosome remodelers and histones (*Figure 9*), although more detailed structural investigation is necessary to elucidate the extent to which these architectures are conserved or distinct.

In summary, we provide here a first structural framework for the interaction of a Swi2/Snf2 ATPase in complex with its protein:DNA substrate. Our structural analyses suggest a two-step mechanism for the remodeling of TBP:DNA:NC2 by Mot1 that could be relevant for other remodelers that act on histone fold protein substrates. It seems plausible to reason that in the first step, TBP's and NC2's interaction with DNA is destabilized by Mot1 binding in an ATP-independent manner. ATP binding does not result in the disassembly of the complex. The formation of the Mot1[NTD]:NC2 clamp ensures robust dependence on ATP hydrolysis, which triggers the final dissociation step, and which most probably occurs by very short-range translocation of the Swi2/Snf2 domain along the minor groove. Considering the relative orientation of Mot1[CTD] and DNA in our model as well as the 3′–5′ tracking

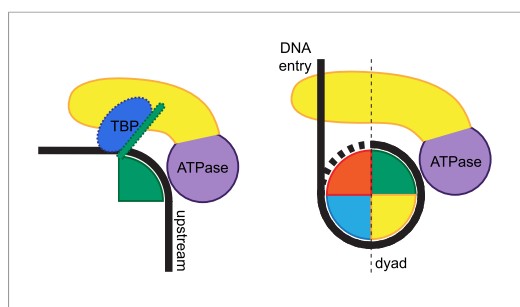

**Figure 9**. Remodeling of substrate protein:DNA complexes by Mot1 and comparison to the ISWI-type nucleosome remodeler. Left side: the binding of Mot1NTD (yellow) induces destabilization of TBP:DNA and NC2: DNA interactions (TBP is shown in blue, NC2 is represented by a green shape). Right side: an ISWI-type Swi2/Snf2 remodeler bound to nucleosome according to current models (*Dang and Bartholomew, 2007*; *Yamada et al., 2011*; *Hota et al., 2013*). The DNA-binding domain (yellow) engages extranucleosomal DNA at the entry site. In both cases, the Swi2/Snf2 ATPase domains (purple) specifically recognize their histone fold: DNA substrates.

direction shown for other Swi2/Snf2 enzymes (*Whitehouse et al., 2003*; *Zofall et al., 2006*), Mot1 may move along upstream DNA towards TBP, disrupting the TBP:promoter interaction.

## Materials and methods

### Protein expression and purification

N-terminally His$_6$-tagged full-length *Encephalitozoon cuniculi* Mot1 (1–1256) and Mot1NTD (1–778) and N-terminally His$_6$-tagged full-length TBP (1–198) were expressed and purified as previously described (*Wollmann et al., 2011*). *E. coli* BL21 Rosetta(DE3) cells (Novagen, Germany) were used to co-express His$_6$-tagged full-length NC2β (1–147) with the untagged full-length NC2α (1–95). Initial purification of the NC2α:NC2β heterodimer was performed using Ni$^{2+}$-NTA resin (Qiagen, Germany). Tobacco etch virus protease digestion was performed to cutoff the tag. Next, ion exchange chromatography using HiTrap SP HP column (GE Healthcare, Germany) connected to AEKTA purifier (GE Healthcare) was applied followed by gel filtration using S75 16/60 column (GE Healthcare) as the final step.

### Selenomethionine labeling

Selenomethionine labeling in insect cells was performed for the N-terminally His$_6$-tagged Mot1NTD construct. High 5 insect cell culture (Invitrogen, Germany) was adapted to Sf-900 II SFM medium (Gibco, Germany) by growing the cells from a starting concentration of $4·10^5$/ml for 4 days (27.5°C, 95 rpm). Subsequently, the cells were diluted to $1·10^6$/ml in 0.5 L of Sf-900 II SFM medium and infected 1: 750 (vol/vol) with P2 virus. The culture was grown for 12 hr and centrifuged (800 rpm, 10 min). The cell pellet was resuspended in 0.5 L of Sf-900 II SFM medium supplied with 75 mg of L-cysteine (Sigma-Aldrich, Germany). After 4 hr of methionine depletion the cells were centrifuged and resuspended in 0.5 L of Sf-900 II SFM medium supplied with 75 mg of L-cysteine and 35 mg of L-selenomethionine (Acros, Germany). The expression was carried out for 48 hr (27.5°C, 95 rpm). The media were supplied with 1.4 µg/mL gentamycin (Roth, Germany) and 10 mM L-glutamine (Gibco).

For the derivatization in *Escherichia coli* expression system, the plasmids were transformed into *E. coli* Rosetta B834 cells (Novagen, Germany) and grown in LB medium (37°C, 200 rpm). The expression cultures were grown in Selenomethionine Medium Base supplemented with Nutrient Mix (Molecular Dimensions, UK). Additionally, selenomethionine solution (Acros) at a final concentration of 42 µg/ml was added to the sterile medium prior to inoculation. The media and selenomethionine solution were prepared according to the instruction supplied by the manufacturer. The cultures were grown until OD$_{600}$ reached 0.4 (160 rpm, 37°C). Next, the temperature was set to 18°C and the cultures were further grown until OD$_{600}$ ≈ 0.7. The expression was induced with IPTG (Roth, 0.5 mM) and carried out overnight (18°C, 160 rpm). All of the used media were supplemented with appropriate antibiotics dependent on the resistance-coding expression plasmids. The applied protein purification protocols did not differ from the ones used for the purification of native proteins.

### Electrophoretic mobility shift assays

Electrophoretic mobility shift assays (EMSAs) were performed essentially as previously described (*Wollmann et al., 2011*) using 47 bp oligonucleotide duplex probes labeled with fluorescein at the 5′ end of one strand. For the EMSAs in *Figure 2*, the top strand DNA sequence was 5′–GGGTAC GGCCGGGCGCCCCGGATGGGGGGCTATAAAAGGGGGTGGGC–3′. Fluorescently labeled DNA (0.5 nM) was incubated for 20 min with TBP (20 nM) and Mot1 or Mot1$^{Δlatch}$ (30 nM) and with or without 30 nM NC2 as indicated. ATP (50 µM) was then added for 5 min. Reactions were incubated at 25°C in 4% glycerol, 4 mM Tris–HCl (pH 8), 60 mM KCl, 5 mM MgCl$_2$, and 100 mg/ml bovine serum

albumin and were resolved on 6% non-denaturing gels. The gels were imaged using a Typhoon Trio phosphorimager as previously described (*Wollmann et al., 2011*). DNA probes used for anti-digoxygenin conjugation (*Figure 8*) were 47 base pairs with the same sequence as above, but with either digoxygenin or fluorescein conjugated to the 5′ end of either the top or bottom strand in each probe. EMSAs performed using the digoxygenin probes and anti-digoxygenin antibody (2.5 nM) were performed in the same was as described above except that reaction products were resolved on 5% gels to improve resolution. Anti-digoxygenin antibody was from Abcam (UK, 21H8).

## Complex formation in gel filtration

DNA and single protein components were mixed and incubated in a stepwise manner at 4°C. TBP was first added to the TATA box-containing DNA in excess. This was followed by the addition of NC2 and Mot1/Mot1$^{NTD}$. Finally, the sample was centrifuged and loaded onto a S200 10/300 GL column connected to AEKTA purifier (GE Healthcare). 20 mM MES pH 6.5, 60 mM KCl, 5 mM MgCl$_2$ and 2 mM DTT or 20 mM HEPES pH 8.2, 60 mM KCl, 5 mM MgCl$_2$ and 2 mM DTT were used for the crystallization or EM and CX-MS analyses, respectively. Oligonucleotides were ordered from Biomers, Germany.

## Crystallization, data collection, and structure determination

For the crystallization of the Mot1$^{NTD}$:TBP:DNA:NC2 complex, 24 double-stranded DNA (5′-AGTA GGGCTATAAAAGGGGGTGGC-3′ top strand) was used. The peak gel filtration fractions were pooled, concentrated by ultrafiltration (Millipore, Germany), and centrifuged. The crystals were grown at 20°C for 7–14 days by hanging drop vapor diffusion technique. Best diffraction quality crystals were obtained by serial streak seeding from selenomethionine-derivatized proteins and grown in 0.2 M imidazole malate pH 5.1 and 9–16% PEG4000 condition. The crystals were flash-frozen in liquid nitrogen using original condition supplemented with 25% glycerol for cryoprotectection. Diffraction data was collected at the *European Synchrotron Radiation Facility* (ID-29) at 100 K and λ = 0.9794 Å. The data were processed with *XDS* (*Kabsch, 2010*) in the space group C 1 2 1 (a = 150.6 Å, b = 140.3 Å, c = 90.8 Å, α = 90.0°, β = 113.7°, γ = 90.0°) with 60% solvent and one complex per asymmetric unit. The structure was successfully solved by molecular replacement with *Phaser*, part of the *CCP4* software suite (*McCoy et al., 2007*; *Winn et al., 2011*). Homology model of the NC2 heterodimer was prepared using *CHAINSAW* (*CCP4*, *Stein, 2008*; *Winn et al., 2011*). The structure was refined in *BUSTER* (v. 2.10.1) at 3.8 Å using TLS refinement strategy (*Bricogne et al., 2011*) and manually rebuilt in *Coot* (*Emsley et al., 2010*). Solvent flattening was performed with *Parrot* (*CCP4*, *Cowtan, 2010*; *Winn et al., 2011*). *B*-factor sharpening and calculation of feature-enhanced 2$F_o$ − $F_c$ map was performed using *PHENIX* (*Adams et al., 2010*; *Afonine et al., 2015*).The histone fold region was characterized by relatively high *B*-factors and poor density and, therefore, the side chains of the residues NC2α 15–89 and NC2β 12–101 were omitted in the final model. The quality of the structure was evaluated with MolProbity (*Davis et al., 2007*); 96%, 4%, and 0.% of the residues were in Ramachandran favored, allowed, and outlier regions, respectively.

## Negative stain EM

For the negative stain EM analysis of the Mot1:TBP:DNA:NC2 complex, 38 dsDNA was used (5′–CAGGCCGGGCGCCCGGCATGGCGGCCTATAAAAGGGTC–3′ top strand). The sample was supplied with 1 mM ADP·BeF$_X$. For the grid preparation sample was diluted to 25 μg/ml. The sample was applied to glow-discharged continuous carbon-coated grids. 5 μl of the sample was incubated for 1 min, blotted dry, washed twice with water, and finally fixed in a 1% (wt/vol) uranyl acetate solution. A total of 144 micrographs were recorded manually on a CM200 field emission gun transmission electron microscope (Philips, Netherlands) operated at 200 keV. Images were recorded on a 4k × 4k Gatan Ultrascan CCD camera using a defocus ranging from 0.5 to 1.5 μm and a dose of 25 e$^−$/Å$^2$. The pixel size corresponded to 1.61 Å on the specimen level. Contrast-transfer function (CTF) determination and phase correction of the micrographs were performed using the TOM software package (*Nickell et al., 2005*). From the corrected micrographs a total of 27,276 particles were selected using the 'boxer' tool from EMAN2 (*Tang et al., 2007*). All subsequent image processing was carried out in XMIPP (*Scheres et al., 2008*). The particles were normalized before further analysis by two-dimensional (2D) reference-free alignment and classification using the ML2D algorithm. Initial

reconstruction of a 3D volume from the particles was carried out using a reference-free approach using a sphere with the approximate particle diameter (10 nm) as an initial model for de novo ML3D alignment. Subsequently, ML3D classification was performed to sort out broken complexes or other non-particles, resulting in 8192 particles used for further refinement. The generated de novo model was filtered to 40 Å and used as an initial reference for iterative projection-matching refinement yielding the final reconstruction of the Mot1:TBP:DNA:NC2:ADP·BeF$_X$ complex at a resolution of 22 Å, according to FSC (0.5 criterion) of two averages, each comprising 50% of the data. The resulting volume showed a good agreement with the previously RCT reconstructed map of Mot1, although more details were present. The map shows all features expected for the determined resolution, most pronounced in the C-shaped contour of the Mot1 protein itself. The resulting volume was furthermore assessed using different approaches (*Figure 7—figure supplement 1*).

## Chemical cross-linking and enrichment of cross-linked peptides

For the CX-MS analysis of the Mot1:TBP:DNA:NC2 complex, 42 double-stranded DNA was used (5′–CAGTACGGCCGGGCGCCCGGCATGGCGGCCTATAAAAGGGTC–3′ top strand). The complex sample at 0.66 mg/ml was supplemented with 1 mM ATPγS, 1 mM ADP, or 1 mM ADP·BeF$_X$ (formed by mixing ADP, BeCl$_2$ and NaF in 1:1:4 molar ratio). An equimolar mixture of isotopically light (d0)- and heavy (d6)-labeled disulfosuccinimidyl glutarate (DSSG; Creative Molecules, Canada) was dissolved in H$_2$O at a concentration of 50 mM. 55 µg of protein complex were incubated with DSSG (0.32 mM final concentration) for 35 min at 30°C (1000 rpm). The reaction was quenched by adding 1 M Tris–HCl pH 8.0 to a final concentration of 100 mM, followed by incubation for further 15 min at 30°C (1000 rpm). The cross-linking efficiency was visualized by SDS-PAGE in combination with silver staining following standard protocols (*Figure 6—figure supplement 1*). Proteins were in the following digested using a standard in-solution protocol. In brief, proteins were denatured by adding two volumes of 8 M urea (Sigma-Aldrich). Cross-linked proteins were reduced with 5 mM final concentration tris(2-carboxyethyl)phosphine (TCEP, Thermo Scientific, Germany) for 45 min at 35°C and subsequently alkylated in the dark for 30 min at room temperature (10 mM iodoacetamide final concentration). Proteins were pre-digested for 2 hr with lysyl endopeptidase (LysC, Wako, Germany) at 35°C at an enzyme-substrate ratio of 1–50 (wt/wt). The protein solution was diluted with four volumes of 50 mM ammonium bicarbonate (ABC), and a second digest was performed overnight with trypsin (Promega, Germany, 1/50 [wt/wt] at 35°C, 1000 rpm). Peptides were acidified with 1% (vol/vol) trifluoroacetic acid (TFA, Sigma–Aldrich) and purified by solid-phase extraction using C18 cartridges (Sep-Pak, Waters, Germany). The desalted eluate was dried by vacuum centrifugation and reconstituted in 20 µl of size exclusion chromatography mobile phase (25% acetonitrile [ACN], 0.1% TFA). 15 µl hereof was injected into a GE Healthcare ÄKTAmicro chromatography system via autosampler. Peptides were separated on a Superdex Peptide PC 3.2/30 column (GE Healthcare) at a flow rate of 25 µl/min. 100 µl fractions were collected in a 96-well plate over a separation window of one column volume. The four fractions containing cross-linked peptides were dried to completeness and reconstituted in 2% ACN, 0.1% formic acid (FA).

## Mass spectrometric analysis of cross-linked peptides and data analysis

Peptides sample were analyzed on an LC-MS/MS system using an UHPLC (EASY-nLC 1000) online coupled to an LTQ Orbitrap Elite (both Thermo Scientific) equipped with a standard nanoelectrospray source. A volume corresponding to an estimated 1 µg of peptide was injected onto a 15 cm × 0.050 mm I.D. reversed phase column packed with 2 µm C18 beads (Acclaim PepMap RSLC analytical column, Thermo Scientific). Peptides were separated using a 60 min gradient of solvent B (98% ACN, 0.1% FA) from 2% to 35% at a flow rate of 250 nl/min. Each sample was injected twice to improve identification of cross-linked peptides. The mass spectrometer was operated in data-dependent mode, selecting up to 10 precursors from a MS$^1$ scan (resolution = 120,000) in a mass range of 300–2000 m/z for rapid collision-induced dissociation (rCID). Singly and doubly charged precursors as well as precursors of unknown charge state were rejected for MS$^2$ selection. CID was performed for 10 ms using 35% normalized collision energy and an activation q of 0.25. Dynamic exclusion was activated with a repeat count of 1, exclusion duration 30 s at a list size of 500 and a mass window of ± 10 ppm. Ion target values were 1,000,000 (or maximum fill time of 10 ms) for the survey scan and 10,000 (or maximum fill time of 100 ms) for the MS$^2$ scan, respectively.

For data analysis, Thermo Xcalibur .RAW files were first converted to the open mzXML format using msconvert tool (ProteoWizard, *Kessner et al., 2008*). Cross-linked peptide candidates were extracted using the xQUEST (v. 2.1.1) pipeline including xProphet for FDR calculations (*Walzthoeni et al., 2012*). Standard settings were used. In brief, data were searched against a self-defined database containing Mot1, TBP, and NC2 proteins. Maximum number of missed cleavages (excluding the cross-linking site) = 2, peptide length = 4–45, enzyme = trypsin, fixed modifications = carbamidomethyl-Cys (57.02146 Da), variable modification = Met-oxidation (15.99491 Da), mass shift of the light cross-linker (96.02112937 Da), $MS^1$ tolerance = 10 ppm, $MS^2$ tolerance = 0.2 for common ions and 0.3 Da for cross-linked ions. The theoretical candidate spectra were scored according to their quality of the match and cross-linked candidates were filtered by a $MS^1$ mass tolerance of −5 to 5 ppm and a Δscore of ≥15% (indicating the relative score difference to the next ranked match). All spectra passing the filtering criteria were further manually validated. Identifications were only considered for the final result list in case both peptides had at least four bond cleavages in total or three adjacent ones and a minimum length of five amino acids. Full list of the detected cross-links can be found in *Table 2—source data 1*.

### Rigid body fitting into EM density map

The rigid body docking of $Mot1^{NTD}$:TBP:DNA:NC2 crystal structure was performed using *colores* applying a 10° sampling step size (*Chacón and Wriggers, 2002*). The search probe was down-filtered to 22 Å, and Laplacian filter (maximizing the fitting contrast) was applied. The DNA was omitted from the model, and the missing side chains of $NC2^{HF}$ were included and set to most common rotamers. The fitting was preformed for the correct and mirrored reconstruction and resulted in several fits of the $Mot1^{NTD}$:TBP:NC2 module. After manual inspection, two very similar solutions (for the rightful hand) were qualified as the correct solution. The other fits were nonsense fits resulted from template drifting, which often occurs at low resolution when the structure represents only a part of the density it is docked into (*Chacón and Wriggers, 2002*).

### Mapping and scoring of the cross links

The set of $Mot1^{CTD}$ orientations (n = 20,000) was generated using *RANCH*, part of the EOM package (*Petoukhov et al., 2012*) in a 'compact' mode, where $Mot1^{NTD}$:TBP:NC2 module was assumed to be flexibly linked to $Mot1^{CTD}$ via the eight amino acid linker (residues 779–786, not present in the atomic models). Next, the models were displayed and the distances were measured using a standard script in PyMOL Molecular Graphics System (v. 1.5.0.4 Schrödinger, LLC). Finally, the models were scored according to the number of cross-links violating the 30 Å cutoff distance. For residues NC2α K61, K62, K92, and NC2β K27, which were not visible in the crystal structure (i.e., comprising short loop regions) but important for the analysis, the distances refer to the position which could be unambiguously modeled based on the crystal structure of NF-Y complex (PDB-ID 4AWL, *Nardini et al., 2013*). All reported distances between detected cross-linked lysines refer to the Euclidean $C_\alpha$–$C_\alpha$ distances between the residues.

### Figure preparation

The figures were prepared using UCSF Chimera package (*Pettersen et al., 2004*), PyMOL Molecular Graphics System (v. 1.5.0.4 Schrödinger, LLC) and OriginPro 8 G (OriginLab, Northampton, MA).

### Accession NUMBERS

The coordinates and structure factors of the crystal structure were deposited in the Protein Data Bank under accession code 4WZS. The EM reconstruction map was deposited in the EMDB database under accession code 2828.

## Acknowledgements

We thank Sebastian Eustermann for the discussion and Oana Mihalache for help with preparation of the grids, the Max-Planck Crystallization Facility (Martinsried) for crystallization trials and the synchrotron facilities ESRF (Grenoble), SLS (Villingen), and DESY-Petra III (Hamburg) for beam time and excellent on-site support. AB acknowledges support from the International Max Planck Research School for Molecular and Cellular Life Sciences.

# Additional information

## Funding

| Funder | Grant reference | Author |
|---|---|---|
| Deutsche Forschungsgemeinschaft (DFG) | SFB 646 | Karl-Peter Hopfner |
| European Research Council (ERC) | ATMMACHINE | Karl-Peter Hopfner |
| National Institutes of Health (NIH) | GM55763 | David T Auble |
| Deutsche Forschungsgemeinschaft (DFG) | GRK1721 | Friedrich Förster, Karl-Peter Hopfner |

The funders had no role in study design, data collection and interpretation, or the decision to submit the work for publication.

## Author contributions

AB, Acquisition of data, Analysis and interpretation of data, Drafting or revising the article; JMS, GS, PR-W, Acquisition of data, Analysis and interpretation of data; FF, Analysis and interpretation of data, Drafting or revising the article; DTA, Conception and design, Acquisition of data, Analysis and interpretation of data, Drafting or revising the article; K-PH, Conception and design, Analysis and interpretation of data, Drafting or revising the article

## Author ORCIDs

Karl-Peter Hopfner, http://orcid.org/0000-0002-4528-8357

# Additional files

## Major datasets

The following datasets were generated:

| Author(s) | Year | Dataset title | Dataset ID and/or URL | Database, license, and accessibility information |
|---|---|---|---|---|
| Butryn A, Schuller JM, Stoehr G, Runge-Wollmann P, Förster F, Auble DT, Hopfner KP | 2015 | Structural basis for recognition and remodeling of the TBP:DNA:NC2 complex by Mot1 | http://www.rcsb.org/pdb/explore/explore.do?structureId=4WZS | Publicly available at the Protein Data Bank (accession no. 4WZS). |
| Butryn A, Schuller JM, Stoehr G, Runge-Wollmann P, Förster F, Auble DT, Hopfner KP | 2015 | Structure of Mot1 in complex with TBP, NC2 and DNA | http://www.ebi.ac.uk/pdbe/entry/emdb/EMD-2828 | Publicly available at EMDB (accession no. 2828). |

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
