## [Decision Letter]

Thank you for sending your work entitled “Structural basis for recognition and remodeling of the TBP:DNA:NC2 complex by Mot1” for consideration at *eLife*. Your article has been favorably evaluated by John Kuriyan (Senior Editor) and three reviewers, one of whom, Gregory D. Bowman, has agreed to share his identity.

SWI/SNF family ATPases play diverse cellular roles, with different classes specialized for different substrates and outcomes. A key aspect of transcriptional regulation and activation is recycling of TATA binding protein (TBP). This recycling is stimulated by the Snf2-type ATPase Mot1. This study builds on previous work by the authors in which they crystallized the HEAT repeat containing N-terminal domain of Mot1 bound to TBP alone. The authors refer to this as representing the “product state”. In this work, to characterize the “substrate state”, the authors solve the crystal structure of the Mot1 N-terminal domain bound to a complex of TBP, DNA and the transcriptional regulator NC2. They provide complementary structural and biochemical analysis to generate a model for the complete Mot1 protein bound to TBP-NC2-DNA. Based on cross-linking mass-spec data and negative stain EM data on full-length Mot1 bound to TBP–DNA–NC2, the authors suggest how the ATPase domain may engage the substrate. Biochemical data are presented to assess how the presence of NC2 affects Mot1 function. The authors present a model in which Mot1 binding to TBP-DNA-NC2 causes an ATP-independent distortion of the DNA that poises the complex for further ATP-dependent destabilization. In this model, interactions between Mot1 and NC2 are coupled to the DNA distortion and the latch domain of the Mot1 N-terminus binds to the DNA-binding surface of TBP to promote full dissociation of TBP from the DNA. The authors further speculate based on cross-linking data that the RecA2-like subdomain of Mot1 may directly affect NC2-DNA contacts.

Overall the work is of high quality and provides new insights into the mechanism of action of Mot1, as well as new structural models for how transcriptional regulators like NC2 affect Mot1 action. In addition, given that NC2 has histone folds, there are potential parallels to nucleosome remodeling enzymes. Specifically, the authors suggest that interactions between NC2 and the Mot1 N-term observed in the crystal structure and the proposed interactions between the RecA2-like lobe and NC2 HF region inferred from cross-linking data may, together, provide a model for how nucleosome remodelers disrupt histone-DNA interactions.

Major points that must be addressed:

1) The fitting of cross-linking data to the ATPase motor modeled in different conformations is used to provide evidence for nucleotide dependent orientations of the ATPase domains. This is interpreted as providing evidence for the “open” conformation in the presence of ADP, as well as a “closed” conformation with ATP-mimics. From the data presented, these conformations appear to be under-determined. In particular, the “open” conformation is unusual, so determining that it is present in the context of a Mot1–TBP–DNA–NC2 complex is potentially an important finding. However, the data to support this more controversial aspect are not compelling. Either this conclusion should be dropped, or a detailed map of cross-links that are gained and lost in support of this large conformational change should be provided. Likewise, there do not appear to be great differences between cross-links of the “semi-closed” and “closed” modeled states. While it is expected that ATP-mimics will stabilize a tightly closed state, the support for one state over the other requires more convincing evidence from cross-linking data to make these claims.

2) The reviewers are concerned that the model presented by the authors may have the direction of translocation backwards. Based on 3' to 5' directionality, the Snf2 insertions should contact the DNA downstream (in front of, as for the location of helicase “pins”) of the contacts conserved among SF2-type helicases. Since the authors orient the insertions toward TBP, it is likely that TBP would be “pushed” rather than “pulled”, assuming that TBP translocates before disruption. Thus, the experiment to test pulling TBP off the DNA ends (with the digoxygenin-Antibody block), shown in Figure 8, doesn't make sense. The authors should clarify this point.

3) If the Mot1-based disruption is the same with or without the histone-fold NC2 domains, then this interaction is presumably not critical. While it is fine to speculate on the parallel between the Mot1/NC2-DNA complex and the interaction of chromatin remodelers with nucleosomal DNA, the conclusions are overstated. The authors could test the functional significance of these interactions (between the insertion domain and NC2-*H5*) by mutagenesis. Do mutations in these regions affect the stability of the complex an/or also the rate of remodeling? If the authors feel that such experiments are beyond the scope of this work then they should tone down the claims regarding the parallel to nucleosomal remodelers.

Other points to address:

1) The authors conclude that the latch is strictly required in the presence of NC2. Quantification of TBP displacement, preferably as a function of time (as in Figure 3), is needed to make this conclusion rigorous.

2) In Figure 2, it will help to label the different bands on the gel.

3) The change in the DNA bend is an important conclusion from this work, but it is difficult to discern in the figures presented. A clearer figure focusing the colors on the DNA bend with and without Mot1 will help.

4) For Figure 3, the reactions appear to be essentially finished after the first time point. Therefore, it would be more appropriate to say that the rate of the reaction is beyond the time resolution of the experiment, whether NC2 is present or not. The authors should not conclude from this that NC2 does not affect the rate.

5) The crystallographic analysis is at quite low resolution (3.9 Å). When the structure is first introduced, an explanation should be provided of the steps taken to verify that the structure is being interpreted accurately.

---

## [Author Response]

*1) The fitting of cross-linking data to the ATPase motor modeled in different conformations is used to provide evidence for nucleotide dependent orientations of the ATPase domains. This is interpreted as providing evidence for the “open” conformation in the presence of ADP, as well as a “closed” conformation with ATP-mimics. From the data presented, these conformations appear to be under-determined. In particular, the “open” conformation is unusual, so determining that it is present in the context of a Mot1-TBP-DNA-NC2 complex is potentially an important finding. However, the data to support this more controversial aspect are not compelling. Either this conclusion should be dropped, or a detailed map of cross-links that are gained and lost in support of this large conformational change should be provided. Likewise, there do not appear to be great differences between cross-links of the “semi-closed” and “closed” modeled states. While it is expected that ATP-mimics will stabilize a tightly closed state, the support for one state over the other requires more convincing evidence from cross-linking data to make these claims*.

We thank the referees for this important comment. We rewrote the relevant paragraph (last paragraph of the subsection “In the presence of ATP analogs Mot1 ATPase domain adopts a ‘closed’ conformation”) and we have additionally included the list of the cross-links between the RecA-like subdomains detected in the presence of different nucleotides (Table 3) together with the distances between the cross-linked sites mapped on the three models, each representing different conformation to address the reviewers’ point. Table 3 shows that apart from the cross-linked sites detected in all three experiments or cross-links that cannot be used to distinguish between different conformations, we also detected several cross-links that were present in one and were absent in the other data sets and for which the distances significantly differ depending on the model used for mapping. All of these cross-links were detected in both the ADP·BeF_x_ and ATPγS data sets or exclusively in the ADP·BeF_x_ data set. Remarkably, all of them are consistent with the “closed” conformation and not with the “open” or “semi-closed”. Thus, our cross-linking data supports the finding that in the presence of ATP-mimicking nucleotides RecA-like subdomains of Mot1^CTD^ adopt “closed” conformation. This conclusion is stated in the revised manuscript. The placement of these cross-links was also depicted in Figure 6. The graphs showing the distance distributions was moved from the old Figure 6 to new Figure 6—figure supplement 2. In Figure 6—figure supplement 2 we showed only the inter-subdomain, and not as before both inter- and intra-subdomain cross-links. Because of the relatively small size of the RecA-like subdomains, with few exceptions all of the cross-links detected within each RecA-like fold lie within the “allowed” distance. We think that this new representation better emphasizes the correlation between the anticipated conformation and the nucleotide state.

The reviewers rightly pointed out that our cross-linking data does not support the finding that in the presence of ADP the Mot1^CTD^ adopts an “open” conformation. Indeed, no cross-links that were unique for the ADP data set were detected. We have dropped the conclusion regarding the “open” conformation and modified this section of the manuscript accordingly (last paragraph of the subsection “In the presence of ATP analogs Mot1 ATPase domain adopts a ‘closed’ conformation”).

Although the differences between the “closed” and “semi-closed” conformations did not appear to be much different in the initial representation, the data included in Table 3 help to distinguish between the fully functional and “semi-closed” autoinhibited state. For this and other reasons we prefer to keep the cross-link distribution mapped on the “semi-closed” conformation represented in Figure 6—figure supplement 2. We are convinced that it will be of interest to readers, since only these three crystal structures of eukaryotic/archaeal Swi2/Snf2 domains are available to date.

*2) The reviewers are concerned that the model presented by the authors may have the direction of translocation backwards. Based on 3' to 5' directionality, the Snf2 insertions should contact the DNA downstream (in front of, as for the location of helicase “pins”) of the contacts conserved among SF2-type helicases. Since the authors orient the insertions toward TBP, it is likely that TBP would be “pushed” rather than “pulled”, assuming that TBP translocates before disruption. Thus, the experiment to test pulling TBP off the DNA ends (with the digoxygenin-Antibody block), shown in*
Figure 8*, doesn't make sense. The authors should clarify this point*.

We thank the reviewers for pointing out our mistake regarding the predicted directionality of the translocation. Indeed, according to the model derived from the CX-MS data, the anticipated Mot1’s movement would likely take place towards in the downstream direction. This mistake was corrected and the Figure 9 was modified accordingly.

Regarding the digoxygenin-antibody block, this is perhaps a misunderstanding. Our dissociation experiments represented by EMSA shown in Figure 8 were designed to distinguish between Mot1 sliding the TBP:NC2 clamp along the DNA over *long distances* using its Swi2/Snf2 translocase domain or alternatively *local* disruption of Mot1:TBP:NC2 complex. The tracking of Mot1:TBP was tested before and it was found that Mot1 does not track (9). However, TBP in complex with NC2 forms a sliding clamp (73), so it is plausible that Mot1:TBP:NC2 tracks, instead of just dissociating. In a physiological context, tracking could redistribute TBP:NC2 in *cis*, while an immediate dissociation could redistribute it in trans (e.g. to different promoters). Our experiment shows that the mechanism employing long-distance processive tracking as shown for chromatin remodeling enzymes (71; 90) is not likely to occur in the Mot1:TBP:NC2 system. We believe that Mot1 disrupts TBP and TBP:NC2 via a non-processive mechanism, e.g. by 1 or a few bp “translocation”, which in principle would be enough to dissociate the protein substrates from a highly constrained DNA. We have used two different DNA substrates that had the antibody block placed on either end to test for both possible tracking directions. We hope that the revised manuscript clarifies this concern.

*3) If the Mot1-based disruption is the same with or without the histone-fold NC2 domains, then this interaction is presumably not critical. While it is fine to speculate on the parallel between the Mot1/NC2-DNA complex and the interaction of chromatin remodelers with nucleosomal DNA, the conclusions are overstated. The authors could test the functional significance of these interactions (between the insertion domain and NC2-*H5*) by mutagenesis. Do mutations in these regions affect the stability of the complex an/or also the rate of remodeling? If the authors feel that such experiments are beyond the scope of this work then they should tone down the claims regarding the parallel to nucleosomal remodelers*.

We thank the referees for pointing this out. We indeed think that analysis of mutants would require in vitro as well as in vivo assays and are beyond the scope of this work. As suggested, we toned down the claims regarding the parallel to nucleosome remodelers.

*Other points to address*:

*1) The authors conclude that the latch is strictly required in the presence of NC2. Quantification of TBP displacement, preferably as a function of time (as in*
Figure 3*), is needed to make this conclusion rigorous*.

We thank the reviewers for suggesting this experiment. We performed kinetic analyses as in Figure 3 but using the Mot1^∆latch^ protein. Fully consistent with our prior work (92), we found that Mot1^∆latch^ was impaired in dissociation of TBP–DNA, but not completely dead catalytically. We found that Mot1^∆latch^ was also defective for dissociation of TBP:DNA:NC2 complexes, but in contrast to the steady-state distribution of complexes after long ATP incubation times, there was no significant difference in the catalytic activity of Mot1^∆latch^ for TBP:DNA vs TBP:DNA:NC2 complexes in this kinetic assay. It was clear from the EMSA data in the original submission that such a differential effect was at best modest; in the revised manuscript we state that the requirement for the latch is no different using complexes with and without NC2 (“The latch was important for this process in reactions both with and without NC2 (Figure 2)”), and we have included the new kinetic data in Figure 3.

*2) In*
Figure 2*, it will help to label the different bands on the gel*.

We added the labels to the Figure 2.

*3) The change in the DNA bend is an important conclusion from this work, but it it is difficult to discern in the figures presented. A clearer figure focusing the colors on the DNA bend with and without Mot1 will help*.

We have modified Figure 5 and Figure 5—figure supplement 1 to better present the DNA conformation. The new Figure 5 shows the comparison of the DNA bound to TBP in the presence and absence of Mot1^NTD^ from TBP’s bottom (concave) side. We have moved the old panel C from Figure 5 to the Figure 5—figure supplement 1. Where our structure is superimposed, we changed the color of DNA from gray to black in order to increase the contrast (Figure 5, and Figure 5—figure supplement 1). We hope that the new representation better reflects the differences between the DNA conformations of the compared states.

*4) For*
Figure 3*, the reactions appear to be essentially finished after the first time point. Therefore, it would be more appropriate to say that the rate of the reaction is beyond the time resolution of the experiment, whether NC2 is present or not. The authors should not conclude from this that NC2 does not affect the rate*.

We fully agree with the reviewers’ opinion. We have modified this statement in the text accordingly (“However, we were not able to detect any significant changes in the Mot1-catalyzed TBP:DNA dissociation rate or the ATP hydrolysis rate of this process in the presence of NC2 (Figure 3)”).

*5) The crystallographic analysis is at quite low resolution (3.9 Å). When the structure is first introduced, an explanation should be provided of the steps taken to verify that the structure is being interpreted accurately*.

We have added the readapted part of the Materials and methods section describing the structure solution and interpretation to the Results section (subsection “Organization of the Mot1^NTD^:TBP:DNA:NC2 complex reveals a high level of structural conservation.”). In particular, we mention in the main text that chain tracing is validated by the anomalous signal of the selenium atoms. After the manuscript was submitted we further improved our model. We noticed some radiation damage in the later frames and could process the data to higher resolution (from 3.9 to 3.8 Å) improving also data and model statistics: significantly lowered R_meas_, at the same *I*/σ*I* and completeness level (see updated Table 1). We then re-refined our model against the revisited data, which lowered R_free_ calculated against copied and extended R_free_ set. Importantly, this did not affect the interpretation of the model.